# Identification of Stable Meta-QTLs and Candidate Genes Underlying Fiber Quality and Agronomic Traits in Cotton

**DOI:** 10.3390/plants14213252

**Published:** 2025-10-24

**Authors:** Abdulqahhor Kh. Toshpulatov, Ozod S. Turaev, Abdulloh A. Iskandarov, Kuvandik K. Khalikov, Sevara K. Arslanova, Asiya K. Safiullina, Mukhlisa K. Kudratova, Barno B. Oripova, Feruza U. Rafieva, Madina D. Kholova, Dilrabo K. Ernazarova, Davron M. Kodirov, Bunyod M. Gapparov, Doniyor J. Komilov, Marguba A. Togaeva, Abduburkhan K. Kurbanov, Doston Sh. Erjigitov, Mukhammad T. Khidirov, John Z. Yu, Fakhriddin N. Kushanov

**Affiliations:** 1Institute of Genetics and Plant Experimental Biology, Academy of Sciences of the Republic of Uzbekistan, Tashkent 111208, Uzbekistan; toshpolatovabduqahhor78@gmail.com (A.K.T.); ozodturaev@gmail.com (O.S.T.); abdulloxiskandarov4@gmail.com (A.A.I.); quvondiqxaliqov87@gmail.com (K.K.K.); arslanovasevara87@gmail.com (S.K.A.); asiyasafiullina0996@gmail.com (A.K.S.); muhlisaqudratova216@gmail.com (M.K.K.); barnoxonoripova127@gmail.com (B.B.O.); feruzarafiyeva25@gmail.com (F.U.R.); mxolova107@gmail.com (M.D.K.); edilrabo64@gmail.com (D.K.E.); davronqodirov.083@gmail.com (D.M.K.); bunyodgapparov20@gmail.com (B.M.G.); dostonerjigitov68@gmail.com (D.S.E.); khidirov.tursunkilovich@gmail.com (M.T.K.); 2Department of Histology and Medical Biology, Tashkent State Medical University, Tashkent 100109, Uzbekistan; qurbonov53a@gmail.com; 3Research Institute of Plant Genetic Resources, National Center for Knowledge and Innovation in Agriculture, Tashkent 100180, Uzbekistan; 4Department of Biotechnology and Microbiology, National University of Uzbekistan, Tashkent 100174, Uzbekistan; 5Department of Biotechnology, Namangan State University, Uychi Street-316, Namangan 160100, Uzbekistan; dkomilov81@mail.ru; 6Department of General Methodological Sciences, Faculty of Digital Technologies, University of Economics and Pedagogy, Karshi 180100, Uzbekistan; togaevamarguba49@gmail.com; 7United States Department of Agriculture (USDA)-Agricultural Research Service (ARS), Southern Plains Agricultural Research Center, College Station, TX 77845, USA

**Keywords:** meta-QTL, *Gossypium* L., fiber quality, genomic regions, marker-assisted selection, candidate genes

## Abstract

Cotton is a globally important crop, with fiber quality traits governed by complex quantitative trait loci (QTL). However, the utility of QTL data is often limited due to inconsistencies across studies. This study conducted a comprehensive Meta-QTL (MQTL) analysis by integrating 2864 QTLs from 50 independent studies published between 2000 and 2024. Of these, 2162 high-confidence QTLs were projected onto a consensus genetic map using BioMercator V4.2.3, resulting in the identification of 75 MQTLs across the cotton genome. These MQTLs exhibited significantly reduced confidence intervals and enhanced statistical support, with 14 MQTLs reported for the first time. Several MQTLs, including MQTLchr7-1, MQTLchr14-1, and MQTLchr24-1, were identified as stable clusters harboring key fiber quality and stress tolerance traits. Candidate gene analysis within select MQTL regions revealed 75 genes, 38 of which were annotated with significant gene ontology terms related to lignin catabolism, flavin binding, and stress responses. Notably, GhLAC-4, GhCTL2, and UDP-glycosyltransferase 92A1 were highlighted for their potential roles in fiber development and abiotic stress tolerance. These findings provide a refined genomic framework for cotton improvement and offer valuable resources for marker-assisted selection (MAS) and functional genomics aimed at enhancing fiber quality, yield, and stress resilience in cotton breeding programs.

## 1. Introduction

Cotton is a major contributor to the global economy, primarily due to its role as a leading source of natural fiber used in the textile industry [1,2,3]. As in many agricultural crops, the expression of quantitative traits in cotton is influenced by several regions of the genome, and each locus contributes to the variation in the trait. These genomic regions are called Quantitative Trait Loci (QTL) [4]. Recently, there has been a significant increase in research focused on enhancing and stabilizing the quality of cotton fiber, which serves as the main raw material in the industry.

QTL mapping has served as a powerful tool for elucidating the molecular mechanisms underlying fiber quality traits. The development and application of this approach, particularly in marker-assisted selection (MAS), have significantly enhanced the efficiency and precision of breeding programs [5]. Since the introduction of molecular mapping technologies, numerous genetic maps have been constructed, leading to the identification of thousands of QTLs linked to economically important traits [6]. QTL mapping has proven particularly effective in investigating the genetic foundation of fiber quality, a range of morphological and yield traits, as well as resistance to various diseases and environmental stresses [4,6,7]. Several studies have identified QTLs associated with multiple fiber quality traits [8,9,10,11,12,13,14,15,16,17,18,19]. In addition, other researchers have reported QTLs related to yield and various morphological characteristics, alongside fiber quality traits [20,21,22,23,24,25,26,27,28]. QTLs linked to resistance against abiotic stress factors have also been mapped [29,30,31,32,33], while QTLs related to biotic stress resistance have been identified as well [34,35,36]. However, the use of diverse populations, markers, marker densities, and testing environments across different studies limits the consistency, reliability, and applicability of these findings in MAS technology.

To address this challenge, a meta-analysis of QTLs offers a valuable approach. By integrating data from multiple studies, it enables the identification of robust QTL regions, known as Meta-QTLs (MQTLs), with reduced confidence intervals (CIs). This method helps mitigate the impact of study-to-study variations, thereby enhancing the reliability of the results [37,38]. Furthermore, Meta-QTL analysis can uncover loci exhibiting pleiotropic effects, indicating that single genomic region influences multiple traits. This phenomenon can be attributed to a pleiotropic gene or tightly linked genes located within the locus. This is accomplished by identifying QTL-rich regions for specific traits and QTL clusters encompassing multiple traits [39]. MQTL analysis has been successfully employed in several key crops, including maize (*Zea mays*) [40,41], rice (*Oryza sativa*) [42,43], wheat (*Triticum*) [44,45], and common bean (*Phaseolus vulgaris*) [46].

In the field of cotton (*Gossypium* L.), MQTL analysis has been utilized to identify QTL hotspots and clusters associated with fiber quality, drought tolerance, and disease resistance. Said et al. (2013) uncovered QTLs linked to both fiber quality and disease resistance, while Lacape et al. (2010) focused on identifying QTLs related specifically to fiber quality traits. Other studies, such as those by Rong et al. (2007) and Abdelraheem et al. (2017), have investigated QTLs associated with fiber quality, flower and leaf morphology tolerance to abiotic and biotic stresses [6,7,39,47].

This study presents a comprehensive MQTL analysis of 2864 QTLs identified from nearly 50 studies published between 2000 and 2024. The primary objective was to systematically investigate the distribution of QTL clusters across the cotton genome, providing valuable insights into the genetic architecture underlying various cotton traits. By synthesizing data from a wide range of studies, this research aimed to highlight the most consistent and robust QTL regions, which could be leveraged for improving the genetic basis of fiber quality, yield, stress tolerance, and other economically significant traits. The findings from this meta-analysis will contribute to the refinement of breeding strategies, including marker-assisted selection (MAS), and enhance the overall understanding of the complex genetic landscape of cotton crop plant.

## 2. Results

### 2.1. Identification of QTLs and Their Distribution Across the Cotton Genome

A total of 2864 QTLs, identified from approximately 50 studies published between 2000 and 2024, were included in the meta-QTL analysis. These QTLs were associated with various traits, including fiber quality, yield, stress tolerance, and morphological characteristics. Information about the QTLs used in the study, their quantity, population type and number, traits, and reference numbers are described in Table 1. The number of QTLs identified in each study varied from 5 to 280, depending on the population size (96-347 individuals) and type (F_2_—second filial generation, BC_1_F_1_—first filial backcross, BC_1_F_2_—second filial backcross, RIL—recombinant inbred line) and the traits studied. Most of the QTLs were reported for fiber-related characteristics such as fiber strength (FS), fiber length (FL), fiber micronaire (FM) and fiber uniformity (FU) indicating a significant focus on improving fiber quality in cotton.

Out of a total of 2864 QTL collected from 50 independent QTL mapping studies, only 2162 were included on the consensus map. 702 QTLs were not used in MQTL analysis due to low LOD values (less than 2), location uncertainty, and location outside the consensus map. The LOD indicator of the QTLs used in the analysis averaged 4.23, ranging from 2.0 to 37.3. In particular, in 2/3 of all QTLs, the LOD score was higher than 3.0. This is one of the important indicators expressing the level of reliability of QTLs used in the study.

Of the QTLs included in the consensus map, 918 are located in the A genome and 1244 in the D genome, and they are unevenly distributed across chromosomes. The highest number of QTLs was identified on chromosomes D03 (136), A08 (135), A05 (120), and D05 (120), while the lowest number of QTLs was found on chromosomes A06 (40) and A02 (42). The distribution of QTLs across the consensus map is described in detail in Appendix A. In this file, chromosome lengths are expressed in centimorgans, and vertical-colored lines along the chromosomes represent individual QTLs. Specifically, QTLs associated with fiber quality traits are depicted in red, those related to yield traits in green, QTLs linked to morphobiological characteristics in blue, QTLs associated with biotic stress resistance in black, and QTLs related to biochemical and physiological properties in yellow.

### 2.2. QTLs Linked to Various Traits and Their Distribution Across the Cotton Genome

Among the 2162 QTLs considered in this study, 1035 were associated with fiber quality traits. A total of 1035 QTLs linked to 11 different fiber properties (FL, FE, FM, FS, FU, FMAT, FSFI, FSCI, FUHML, FB, FR) were integrated into the consensus map and directly utilized for MQTL analysis. The highest numbers of QTLs were associated with FL, FS, and FM traits, totaling 222, 297, and 191, respectively. While the most QTLs (21) genetically linked to FL were located on chromosome A05, the highest number of QTLs for FS (41) was recorded on chromosome A07, and for FM (20 QTLs) on chromosome D08. These findings highlight chromosomes A05, A06, and D06 as crucial QTL clusters for fiber quality improvement. The concentration of numerous QTLs in these regions suggests the potential presence of major genes or tightly linked gene clusters controlling these economically important traits. Therefore, these genomic regions could be prioritized in targeted genetic selection and marker-assisted selection (MAS) strategies. The number of QTLs influencing FE and FU traits was also relatively high, at 122 and 150, respectively. For other fiber properties (FMAT, FSFI, FSCI, MV, FR), comparatively fewer QTLs were integrated into the consensus map.

Regarding yield-related traits, 437 were directly utilized in the MQTL analysis. These QTLs were associated with 17 different yield properties, with the highest numbers observed for traits such as BW (72), SI (85), and LP (105). QTLs linked to other yield traits were relatively less frequent, varying between 1 and 46. Notable chromosomes for the BW trait were 18 and 21, where 11 and 7 QTLs were recorded, respectively. The highest number of QTLs associated with the SI trait (8 each) were located on chromosomes 3 and 7, while the most QTLs linked to LP (17) were also observed on chromosome 3, similar to SI.

From a total of 475 QTLs associated with 28 different morpho-biological traits represented on the consensus map, 150 belonged to the PH trait. On chromosome 17, the highest number of QTLs (33) affecting plant height was observed. This concentration of QTLs for plant height on chromosome 17 demonstrates its central role in regulating plant architecture. Plant architecture is a key factor for optimizing cultivation practices and facilitating efficient mechanical harvesting in cotton. This genomic region warrants further investigation in cotton breeding programs aimed at developing desirable plant morphology.

Additionally, 134 QTLs associated with physiological and biochemical traits (21 types) and 81 QTLs related to biotic stress resistance were directly used for MQTL analysis. The chromosomal locations and frequencies for all these traits are fully detailed in Appendix A.

### 2.3. Meta-QTLs (QTL Clusters) and Their Distribution Across the Genome

Based on 2162 QTL data points included in the consensus map, MQTL analysis using the Biomercator 4.2 program identified 39 MQTL regions along the A genome and 36 MQTL regions in the D genome. Specifically, 4 MQTLs were detected on each of 10 chromosomes (A01, A03, A11, A05, A08, A09, D02, D10, D11, D06), 3 on each of 7 chromosomes (A10, A12, D03, D04, D05, D08, D12), 2 on each of 6 chromosomes (A04, A06, A07, A13, D01, D09), and 1 on each of 2 chromosomes (A02 and D13). The variation in the number of MQTLs across chromosomes may be more related to the proximity or overlap of QTLs rather than the number of initial QTLs. For instance, although the D07 chromosome encompassed 84 initial QTLs, no MQTL was detected on this chromosome due to the relatively dispersed arrangement of QTLs. In contrast, the A07 chromosome, which contains 111 QTLs, yielded 2 MQTLs, while A11, containing 67 QTLs, produced 4 MQTLs. Similar patterns can be observed in several other chromosomes. Nonetheless, a relatively high number of closely positioned and overlapping QTLs ensures the detection of MQTLs with a shortened CI (Confidence Interval).

During the analysis, only genomic regions containing at least 4 QTLs were recorded as MQTLs. Out of a total of 75 MQTLs, 3 (MQTLchr3-3, MQTLchr3-3, MQTLchr25-3) contain exactly 4 QTLs. MQTLs containing the largest number of QTLs were identified on chromosomes D02 (MQTLchr14-1 with 23 QTLs), A08 (MQTLchr8-1 with 22 QTLs), D01 (MQTLchr15-2 with 19 QTLs), D13 (MQTLchr18-1 with 18 QTLs), and A09 (MQTLchr9-1 with 18 QTLs), with almost 50% of the total QTL clusters containing 10 or more QTLs.

The CI value also decreased several times compared to the initial QTLs, averaging 0.976, while the CI across MQTLs ranged from 0.06 to 3.2. Several QTL clusters, such as MQTLchr7-1, MQTLchr9-1, MQTLchr11-1, MQTLchr14-1, MQTLchr17-1, MQTLchr19-2, and MQTLchr24-1, have a very short CI (<0.5) while containing a relatively large number of QTLs: 15, 18, 14, 23, 15, 14, and 15, respectively. Accordingly, these QTL clusters may be reliable genomic regions with pleiotropic effects. The fact that the locations of these MQTLs also correspond to the results of several previous studies further confirms their significance as regions containing genes that influence the development and variability of fiber quality and various yield characteristics.

The locations of MQTLs identified during the study were compared with 6 previously published MQTL analysis reports [39,47,74,75,76,77]. According to the comparison results, 61 MQTLs correspond to previous reports. 14 out of 75 MQTLs were identified in this study for the first time (MQTLchr1-2, MQTLchr1-4, MQTLchr3-4, MQTLchr4-2, MQTLchr8-4, MQTLchr9-4, MQTLchr14-3, MQTLchr17-2, MQTLchr19-3, MQTLchr20-3, MQTL21-2, MQTL21-4, MQTLchr22-3, MQTLchr25-3). Of the MQTLs recorded on the consensus map, 21 have a similar position to the QTL clusters in only one previously published study, while the locations of 38 MQTLs correspond to the results of at least two studies. Notably, 2 QTL clusters, MQTLchr14-1 and MQTLchr23-1, have similar positions to the QTL clusters in 5 previous studies on the topic, suggesting that these genomic regions may be reliable target areas for identifying candidate genes (see Table 2).

### 2.4. Candidate Gene Identification

Initially, MQTLchr7-1, MQTLchr9-1, MQTLchr11-1, MQTLchr14-1, MQTLchr17-1, MQTLchr19-2, MQTLchr23-1, and MQTLchr24-1 QTL clusters were selected as relatively reliable genomic regions for identifying candidate genes due to their relatively short confidence intervals (CI), coverage of multiple QTLs, and greater consistency with previous studies. The MQTLchr9-1 QTL cluster was not used to identify candidate genes due to the lack of reliable flanking markers for this region. In QTL clusters MQTLchr14-1, MQTLchr17-1, MQTLchr19-2, MQTLchr23-1, and MQTLchr24-1, the physical CI was less than 1 MB, and BLAST 2.13.0+ analyses were performed based on gene sequences located within the CI. As a result, 75 candidate genes were initially identified across these regions (Appendix A).

### 2.5. Gene Ontology and KEGG Analysis

Gene ontology and KEGG (Kyoto Encyclopedia of Genes and Genomes) pathway analysis were performed for these candidate genes using Shiny GO v0.82. Of the initial 75 candidate genes, 38 were annotated with gene ontology (GO) terms, providing valuable insights into their potential biological roles.

The GO enrichment analysis aimed to investigate the biological processes, molecular functions, and cellular components associated with this set of candidate genes. The results of the enrichment analysis revealed significant overrepresentation (*p* < 0.05) of genes linked to several key GO terms, which were categorized into three primary domains: biological process, molecular function, and cellular component.

To strengthen the biological interpretation, enriched GO categories and pathways were explicitly linked to candidate genes and their associated traits. Figure 1 illustrates the most significant GO terms within these three categories, providing a visual representation of the biological relevance of the identified candidate genes. In particular, genes related to lignin metabolism, secondary cell wall biosynthesis, and ROS signaling were strongly associated with fiber strength, elongation, and stress tolerance.

Molecular function: A total of 18 candidate genes were associated with 11 molecular function GO terms. The most significantly enriched terms were: Acetolactate synthase activity (GO: 0003984, Fold Enrichment: 164.2 −log10(FDR): 1.35), Hydroquinone–oxygen oxidoreductase activity (GO:0052716, Fold Enrichment: 39.26, −log10(FDR): 1.77) and Flavin adenine dinucleotide binding (GO:0050660, Fold Enrichment: 17.65 −log10 (FDR): 1.77).

Biological process: In the biological process category, 23 candidate genes were associated with 13 GO terms. The most highly enriched terms in this category are Lignin catabolic process (GO: 0046274, Fold Enrichment: 39.262, −log10(FDR): 1.69).

Cellular component: 8 candidate genes are associated with five cellular component GO terms. Among them, Apoplast (GO:0048046, Fold Enrichment: 21.9, −log10(FDR): 1.72), is enriched with high values (Appendix A).

Based on KEGG pathway enrichment analysis, candidate genes associated with MQTLs (Meta-QTLs) showed significant enrichment primarily in carbon and energy metabolism pathways. The most notable findings include: Butanoate metabolism (FDR = 0.005, fold enrichment = 102.5), Glyoxylate and dicarboxylate metabolism (FDR = 0.01), 2-Oxocarboxylic acid metabolism (FDR = 0.016).

Significant enrichment was also observed in central metabolic pathways such as glycolysis–gluconeogenesis, pyruvate metabolism, and carbon metabolism (FDR < 0.05). The coordinated activation of these pathways is critical for providing energy and carbohydrate resources necessary for fiber cell development and maturation (Appendix A).

Notably, the genes LOC107892107 and LOC107892112 were recurrently implicated across multiple key metabolic pathways, indicating their central role in metabolic processes regulating fiber synthesis and quality.

GhLAC-4 and GhLAC-17 genes, which were predicted to be involved in multiple biological pathways across all three primary GO categories (biological process, molecular function, and cellular component), belong to the multifunctional Laccase family. Laccases are copper-containing enzymes that play critical roles in various physiological processes such as cell elongation, lignification, and pigmentation. These enzymes are considered essential in the polymerization of monolignols into lignin, a key component of the plant cell wall that contributes to the structural integrity of plant tissues [79]. Laccases, often classified as phenyloxidases, are involved in the biosynthesis of lignin, which provides mechanical strength, resistance to pathogens, and contributes to the overall quality of fiber in cotton.

Balasubramanian et al. (2016) studied *G. arboretum* demonstrating that the GaLAC-4 and GaLAC-17 genes are vital for lignin biosynthesis during the formation of secondary cell walls, directly contributing to improved fiber quality and strength [80]. In this study, the GhLAC-4 and GhLAC-17 genes were found to be located within the MQTLchr11-1 QTL cluster region, which encompasses 14 QTLs, seven of which are associated with fiber quality traits. This colocalization suggests that these genes could be important candidates for improving fiber strength and quality in cotton breeding programs.

The UDP-glycosyltransferase 92A1 (LOC107938717) gene, found in the MQTLchr14-1 cluster, may be associated with heat stress tolerance. This conclusion is supported by the research of Li et al. (2024) on the *ZmUGT92A1* gene in maize [81]. The role of the UGT family in responding to abiotic stress by modifying secondary metabolites (Chen et al., 2023) makes this gene a promising candidate for enhancing heat stress tolerance in cotton [82,83].

Three important genes were simultaneously identified in the MQTLchr17-1 region: GhCTL2 (LOC107892108), non-specific lipid transfer protein (LOC107918814) and SAUR36 (LOC121215603). The GhCTL2 gene may play an important role in the formation of secondary cell walls in fibers, particularly influencing the mechanical properties of the fiber by regulating the interaction between cellulose microfibrils and hemicelluloses [84,85]. The lipid transfer protein (LOC107918814) may regulate fiber elongation through phosphatidylinositol transport, according to Deng et al. (2016) [86].

Meanwhile, the SAUR36 gene (LOC121215603) may influence fiber maturation and leaf senescence processes through auxin signaling. The fact that overexpression of the homologous gene in Arabidopsis causes premature senescence suggests that a similar mechanism might operate in cotton. However, due to interspecific differences, further research is needed [87].

The flavin-containing monooxygenase 1 gene (LOC107907082) in the MQTLchr19-2 region may play an important role in plant stress responses. Czarnocka et al. (2015) have shown that this gene enhances tolerance to excessive light stress by regulating reactive oxygen species (ROS) signaling [88]. Zhao et al. (2018) found that FMO genes could regulate the flowering process [89].

Finally, the polygalacturonase gene (LOC107888934), identified in the MQTL23-1 region, may influence fiber cell wall dynamics by breaking down pectins [90]. This gene could play an important role in improving FL and flexibility (FF).

These genes may serve as important genetic markers for improving cotton fiber quality and enhancing abiotic stress tolerance. However, further functional studies are required to elucidate their specific mechanisms and assess the possibilities for practical application. Identifying the specific function and potential applications of each gene could be an important direction for future research.

## 3. Discussion

Meta-QTL analysis serves as a powerful strategy to refine the genetic resolution of QTL mapping by integrating data across independent studies, thus enabling the identification of stable and reliable consensus loci. This approach is particularly effective in elucidating the genetic architecture underlying complex agronomic traits such as fiber quality, yield, disease resistance, and abiotic stress tolerance. By harmonizing diverse QTL datasets, Meta-QTL analysis provides a consolidated framework that enhances both biological interpretation and practical utility in breeding programs.

Several prior studies have demonstrated the effectiveness of MQTL analysis in cotton. For instance, Lacape et al. [7] analyzed over 1200 QTLs derived from RIL and BC populations of *Gossypium hirsutum × G. barbadense*, identifying consensus clusters for fiber length, fineness, and color. Said et al. [39] conducted a meta-analysis on 1223 QTLs from 42 studies and reported 76 MQTLs and approximately 50 QTL hotspots associated with fiber, morphological, and stress-resilience traits. More recently, Yuan et al. [75] identified 84 MQTLs associated with earliness-related traits, further reinforcing the robustness of this method in detecting stable loci across the cotton genome.

In the present study, a total of 2864 QTLs were initially compiled, of which 2162 with high statistical confidence were projected onto a consensus genetic map, leading to the identification of 75 MQTLs. Notably, 14 of these were reported here for the first time. Among these, MQTLchr7-1, MQTLchr11-1, MQTLchr14-1, MQTLchr17-1, MQTLchr19-2, MQTLchr23-1, and MQTLchr24-1 displayed short confidence intervals and high QTL density, making them prime candidates for marker-assisted selection (MAS) and candidate gene discovery.

Gene mining within these regions revealed 75 candidate genes, 38 of which were functionally annotated using Gene Ontology (GO) analysis. For example, GhLAC-4 and GhLAC-17, identified in MQTLchr11-1, encode laccase enzymes involved in lignin polymerization—a key process in secondary cell wall formation, potentially enhancing fiber strength. MQTLchr14-1 harbored the UDP-glycosyltransferase 92A1 (LOC107938717), whose homolog in maize has been linked to improved thermotolerance, suggesting a possible role in heat stress adaptation in cotton.

A particularly compelling result was the co-localization of three functionally distinct genes—GhCTL2, SAUR36, and a lipid transfer protein within MQTLchr17-1. These genes are putatively involved in cell wall assembly, fiber elongation, and auxin-mediated senescence, respectively, indicating a potential regulatory module for fiber development. Similarly, the flavin-containing monooxygenase 1 gene (LOC107907082), found in MQTLchr19-2, has been associated with reactive oxygen species (ROS) regulation and flowering, underscoring its multifunctional role. Furthermore, the polygalacturonase gene (LOC107888934) located in MQTLchr23-1 may modulate pectin degradation and thereby contribute to FF and FE.

While these findings align with and build upon previous reports, it is important to note certain limitations. The accuracy of QTL projection is dependent on the resolution and marker density of the original maps, and although consensus map construction mitigates this to an extent, variability in population type and environment remains a confounding factor. Moreover, while in silico candidate gene prediction provides a useful starting point, functional validation via expression profiling, gene editing (e.g., CRISPR/Cas9), or transgenic approaches is essential to confirm causal relationships. Although a combined MQTL + GWAS approach was not applied in the present study, it is recognized as a valuable strategy to further narrow MQTL intervals and improve the resolution of candidate gene discovery. This approach is therefore highlighted as an important direction for future research.

In conclusion, this study significantly advances our understanding of the genetic basis of fiber quality and other agronomic traits in cotton. The identified MQTLs and candidate genes represent valuable genomic resources for accelerating precision breeding through MAS. Future research should focus on validating gene function across different genetic backgrounds and stress conditions, dissecting gene regulatory networks, and exploring potential pleiotropic interactions to fully harness the genetic potential uncovered through meta-QTL analysis.

## 4. Materials and Methods

### 4.1. Collection of QTL Data

For the Meta-QTL analysis, information on individual QTLs was systematically collected from peer-reviewed publications indexed in Scopus, Web of Science, PubMed, Google Scholar, and ResearchGate databases. A comprehensive literature search was performed using combinations of keywords such as “cotton”, “QTL mapping”, “fiber quality”, “yield”, “stress tolerance”, and “*Gossypium*”.

From each study, the following parameters were extracted: trait name, chromosome number, flanking or nearest marker(s) and their positions, most probable QTL position, mapping population type (e.g., F_2_, BC_1_, RIL) and size, logarithm of odds (LOD) score, and either phenotypic variance explained (PVE) or R^2^ value. When multiple QTLs were reported for the same trait, each was recorded separately.

A total of 2864 QTLs were initially collected from 50 independent studies. Thereafter, a strict filtration protocol was applied to ensure data quality for the meta-QTL analysis. Only QTLs meeting the following criteria were retained:(a)Possession of a well-defined confidence interval (CI) with clear start and stop positions.(b)An LOD score of ≥2.0.(c)The value of phenotypic variation (PVE or R2) is indicated.

QTLs for which only the peak position was provided, without explicit CI boundaries, were excluded from the analysis. In cases where the most likely QTL position was not specified, the midpoint between the start and stop positions was used as the reliable position.

As a result of this rigorous filtering process, a final set of 2162 QTLs was compiled for subsequent analysis.

### 4.2. Software for MQTL Analysis

MQTL analysis was conducted using BioMercator v4.2.3, a specialized platform for integrating QTL data from multiple genetic maps into a consensus map and statistically identifying MQTLs [91]. The software enables projection and alignment of QTLs from different studies by standardizing chromosome designations, flanking marker positions, and mapping population parameters.

To ensure analytical rigor, only QTLs with clear marker information and reliable statistical support (LOD values above the threshold defined in the original studies, and PVE or R^2^ values) were included in the analysis. The software applies the two-step algorithm of Veyrieras et al. (2007), which uses multiple model selection criteria, including the Akaike Information Criterion (AIC), Corrected AIC (AICc), AIC3, Bayesian Information Criterion (BIC), and Average Weight of Evidence (AWE), to estimate the most likely number and positions of MQTLs per chromosome [38].

In addition to its computational framework, the graphical interface of BioMercator facilitated the visualization of QTL distributions, overlapping clusters, and their confidence intervals (CIs). This functionality aided in the accurate identification of consensus genomic regions and provided a reliable basis for downstream applications such as candidate gene mining and marker-assisted selection (MAS).

### 4.3. Consensus Map Construction

A consensus genetic map was constructed using the LPmerge package in R v4.5.1, based on seven high-density individual genetic maps: TM-1 × Hai-7124, BC1 (2008) [92]; Emian-22 × 3-79, BC1 (2011) [93]; Guazuncho-2 × VH8-4602; CCRI 12-4 × (AD) 5-7 [94]; TM-1 × 3-79, RIL [22]; and Yumian-1 × T586, RIL (2015) [13]. The following parameters were applied during the construction:(a)The maximum interval for each linkage group was set to 1:4.(b)The accuracy of potential consensus maps was evaluated by comparing their root mean square error (RMSE) values.(c)The consensus map with the smallest average RMSE value was selected for subsequent analyses.

As a result, a consensus map spanning 3988.9 cM and incorporating 11,370 molecular markers, including SSR (Simple Sequence Repeat) and RFLP (Restriction Fragment Length Polymorphism) markers, was developed.

The distribution of markers across chromosomes and the total genetic length of the consensus map are shown in Figure 2 and Figure 3, respectively.

### 4.4. QTL Projection and MQTL Identification

After rigorous filtering, a total of 2162 QTLs were projected onto the consensus map constructed with LPmerge, visualized using BioMercator v4.2. This procedure accounted for the uniformity of *p*-values of flanking markers as well as the minimum distance ratio between genetic maps. According to standard settings, these parameters were set at 0.25 and 0.50, respectively [75]. The MQTL analysis was performed using the two-step algorithm of Veyrieras et al. (2007) implemented in BioMercator v4.2. Prior to this, the distribution and density of projected QTLs along each chromosome were visually inspected to estimate the approximate number of QTL clusters [38,96]. The first step of the Veyrieras algorithm identified the number of potential MQTL models based on multiple criteria, including the Akaike Information Criterion (AIC), Corrected AIC (AICc), Modified AIC with a coefficient of 3 (AIC3), Bayesian Information Criterion (BIC), and Average Weight of Evidence (AWE). The best MQTL model was selected based on achieving the lowest scores in at least three of these five criteria. The second step required inputting a value between 1 and 10 to specify the number of top MQTL regions to display. Consequently, the cluster numbers estimated during the visual inspection were used as input for the software, enabling the identification of the most probable number and positions of MQTLs on each chromosome. Genomic regions containing four or more overlapping QTLs within a 20 cM interval were defined as MQTLs [39,42]. This two-step approach enabled the identification of consensus QTL regions with high resolution and statistical confidence, thereby establishing a robust foundation for the effective application of marker-assisted selection (MAS) in cotton breeding programs.

This two-step approach allowed the identification of consensus QTL regions with higher resolution and statistical confidence, thereby providing a robust framework for downstream candidate gene discovery and enabling more efficient application of marker-assisted selection (MAS) in cotton breeding programs.

### 4.5. Identification of Candidate Genes

From the total of 75 identified MQTLs, clusters with the shortest confidence intervals (CIs) and the largest number of contributing QTLs were prioritized for candidate gene mining. To determine the physical positions of MQTLs, in silico PCR analysis was performed using the *Gossypium hirsutum* reference genome (*Ghirsutum_578_v3.0.fa*). The analysis was based on the flanking markers of the selected MQTL regions, and the corresponding physical positions (bp) of the markers were identified. The peak position (bp) of each MQTL was then estimated using the following formula [38]:
$$Peak\;position(bp)=starting\;position\;(bp)+\left[\begin{array}{c} \underline{\left(end\;position\;\left(bp\right)-starting\;position\;(bp)\right)}\\ \left(end\;position\;\left(cM\right)-starting\;position\;(cM)\right) \end{array}\right]\times \begin{array}{c} \underline{CI}\\ 2 \end{array}$$

Based on the calculated peak position, a physical interval (bp) was defined for each MQTL. The genes located within these intervals were retrieved from the *G. hirsutum* TM-1 UTX_v3.1 genome assembly available at the CottonGen database.

The nucleotide sequences of the identified genes were further validated and functionally annotated using the NCBI BLASTn tool, with *Gossypium hirsutum* (taxid: 3635) specified as the target organism. This step enabled functional annotation and homology-based characterization of candidate genes potentially associated with fiber quality and key agronomic traits, thereby strengthening the biological relevance of the identified MQTL regions.

### 4.6. Gene Ontology Analysis

Gene Ontology (GO) and KEGG analysis of the identified candidate genes was carried out using the ShinyGO v0.81 web-based platform (https://bioinformatics.sdstate.edu/go/, accessed 12 August 2025). This tool enables large-scale functional annotation by identifying statistically overrepresented GO terms associated with a given gene set.

The analysis classified enriched terms into the three principal GO categories: Biological Process (BP), Molecular Function (MF), and Cellular Component (CC). Enrichment significance was determined using a threshold of *p* < 0.05, and only terms meeting this criterion were considered biologically relevant.

To increase the robustness of interpretation, GO terms were further examined for functional consistency with traits of interest, particularly those related to fiber development, stress tolerance, and yield-associated pathways. This approach allowed the identification of the most biologically meaningful categories, thereby strengthening the link between the identified candidate genes and agronomically important traits in cotton.

## 5. Conclusions

This study provides a significant contribution to cotton genomics by identifying 75 Meta-QTL (MQTL) clusters associated with fiber quality and other key agronomic traits. These MQTLs refine our understanding of the complex genetic architecture underlying fiber strength, length, yield, and stress tolerance. The integration of QTL data from multiple independent studies enables the identification of stable and reliable genomic regions, offering a valuable resource for molecular breeding in cotton.

The candidate genes identified within these MQTL regions reveal critical insights into the molecular mechanisms governing fiber development and stress adaptation. Their functional annotations support their roles in processes such as secondary cell wall biosynthesis, auxin signaling, ROS regulation, and stress-related metabolic pathways. These findings have practical implications for the application of molecular markers and marker-assisted selection (MAS) to accelerate breeding for superior cotton cultivars.

Despite these promising outcomes, further research is essential to validate the function of these candidate genes. Functional genomics approaches, including expression profiling, transgenic validation, and gene editing, will help confirm their regulatory roles and potential utility in breeding programs. Additionally, verifying the stability of these MQTLs across diverse genetic backgrounds and environmental conditions will strengthen their applicability in a wide range of production systems.

In summary, this meta-QTL analysis represents a valuable step toward precision breeding in cotton, enabling the development of high-performing cultivars adapted to future agricultural challenges. These results contribute to the long-term goals of improving cotton productivity, fiber quality, and environmental resilience through genomics-assisted breeding strategies.

## Figures and Tables

**Figure 1 plants-14-03252-f001:**
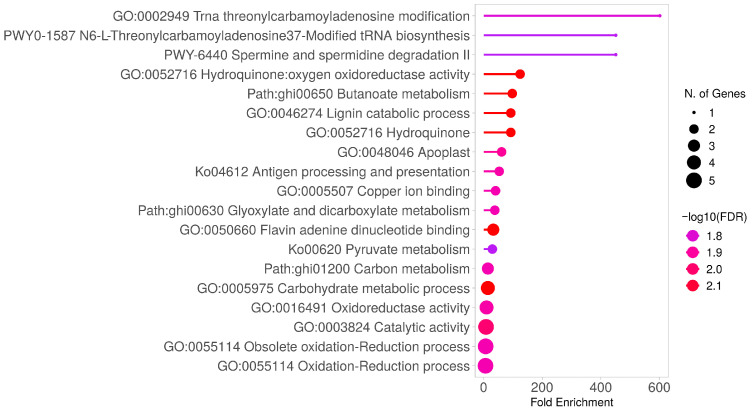
**Functional Annotation and Pathway Analysis of Candidate Genes in Cotton Meta-QTL Regions.** The *X*-axis represents fold enrichment, while the color gradient indicates the −log10(FDR) value, corresponding to the statistical significance of enrichment. Significant enrichment was defined as *p* < 0.05 and −log10(FDR) > 1.3 [78]. **Abbreviations:** MQTL—Meta-QTL; GO—Gene Ontology; FDR—False Discovery Rate.

**Figure 2 plants-14-03252-f002:**
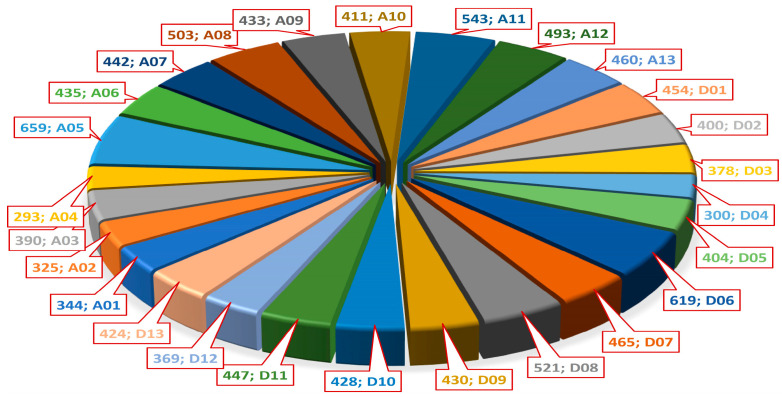
**Chromosome-wise distribution of molecular markers in the consensus genetic map.** Each wedge of the pie chart represents the number of markers assigned to individual chromosomes (e.g., 659 on A05, 503 on A08, 619 on D06, etc.), highlighting marker density across the A and D sub-genomes of cotton. A-subgenome chromosomes are labeled A01–A13, and D-subgenome chromosomes are labeled D01–D13, according to the standard cotton genome nomenclature [95].

**Figure 3 plants-14-03252-f003:**
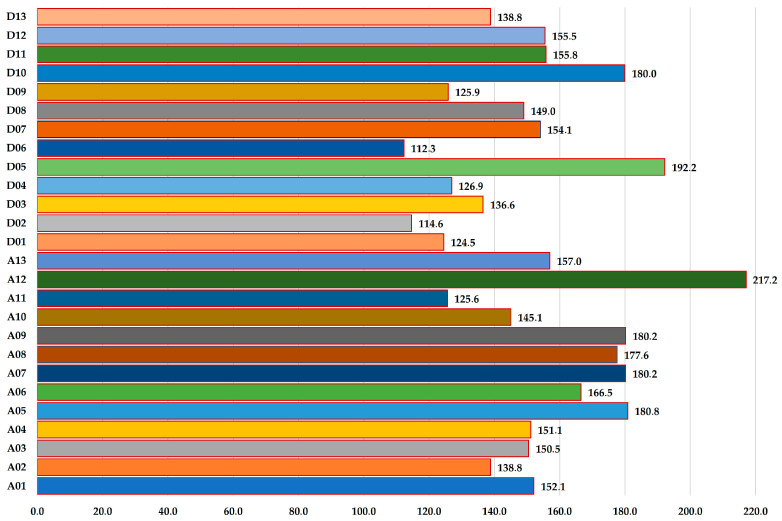
**Total genetic map length (cM) of each chromosome in the consensus map.** The bar chart represents the cumulative genetic distance (in centimorgan, cM) for each of the 26 chromosomes (A01–A13 and D01–D13). Chromosomes A12 (217.2 cM), D05 (192.2 cM), and A08 (180.2 cM) exhibited the longest map length, suggesting extended recombination distances and potentially higher marker density or saturation in these regions. **Abbreviation:** cM—centiMorgan.

**Table 1 plants-14-03252-t001:** Information about scientific publications used in meta-QTL analysis, their authors, populations used, number of QTLs and traits.

No.	Population Type and Size	No. of QTLs	Traits	Ref.
1.	F_2_ (4Su-271; 4I-248; SgJ-276; Sg4-304)	50	PH, BW, LP, FL, FS, FU, FE, FM,	[28]
2.	F_2_, 251	5	NFB	[48]
3.	RIL, 180	113	BW, LP, SI, FL, FU, FM, FE, FS	[26]
4.	F_2,_ 270	79	LP, CP, CO, LA, OA, PA	[23]
5.	RIL, 180	86	LP, SY, SI, BW, FE, FL, FS, FSCI, FBN	[49]
6.	F_2:3_	5	LP	[50]
7.	Ils, 115	60	FL, FS, MIC, FU, FE	[17]
8.	RIL	27	FL, FS, FM	[51]
9.	RIL, 180	33	FL, FS, FM, FMAT, FR, FB, FE, FSFI	[8]
10.	RIL, 180	62	FL, FU, FM, FE, FS	[13]
11.	BC_1_F_2_, 115	44	FL, FU, FM, FE, FS	[14]
12.	Composite cross-population, 172	63	FE, FL, FM, FU, FS	[11]
13.	BC_2_F_1_, 133	153	FL, FS, FM, BW, SI, LI	[27]
14.	RIL, 177	41	PH	[52]
15.	RIL, 180	59	FL, FU, FS, FE, FM	[15]
16.	F_2:3_, 188	11	RL, SFW, RFW, SDW, RDW, CHL, SH	[29]
17.	F_2:3_, 155; RIL, 190	50	FS, FL, FM, FU, FE	[10]
18.	F_2_, 150	15	CL	[35]
19.	RIL, 200	11	LFMP, JI, SCY	[53]
20.	F_2:3_, 229	41	VR	[34]
21.	RIL, 178	134	FL, FU, MIC, FE, FS, SCW, LW, LP, SI, BN	[24]
22.	composite cross-population	11	FL, FS, FU	[9]
23.	F_2_, 124	33	SW, LW, LP, LI, SI, MV, FE, FS, FUHML, HSW	[20]
24.	a four-way cross-mapping population (4WC), 239	74	PB, NB, BW, LP, LI, SI, FL, FS, FM, FU, FE	[21]
25.	RIL, 196	8	Oil, Pro	[54]
26.	RIL, 177	55	FL, FU, FS, FE, FM	[18]
27.	BIL, 146	67	FL, FS, FM, FE, FU, BW, LP, LY, SCY	[55]
28.	F_2:3_	50	SY, LY, LP, BN, BS, LI, SI	[56]
29.	RIL, 177	34	MRL, PH, RL, RSA, NRT, NRF, SW, RW, RV	[57]
30.	F_2_	43	PH, FN, BS, SY, SI, LY	[58]
31.	RIL, 163	9	FOV	[36]
32.	F_2:3_, 173	39	FL, FU, FM, FS, FE	[12]
33.	F_2:3_, 188	8	Bla, Fcc, Bcc, Fbbw	[59]
34.	F2, 347	18	LP, BW, FL, FM, FS	[60]
35.	F_2_, 96	17	FSH, SNH, PBS, TNSa, TNN, NOB, TNB, LOB, LOS, LOp	[61]
36.	F_2_, 123	17	BW, LP, FL, FU, FM, FE	[62]
37.	RIL, 177	24	FL, FS, FU, FE, FM	[19]
38.	F_5_, 122	19	PH, RWC, CSI, PC, TCC, NRA	[63]
39.	RIL, 196	37	FL, FM, FS	[16]
40.	RIL, 186	16	PA, YC, FP	[22]
41.	F_2_, 249	112	PH, CNH, FTLH, STLH, SLA, SPn, TPn, Sci, Tci, Scond, Tcond, STr, TTr, Chla, FBN, SCY, LY, BW, SI, LP, LI, FL, FS, MC	[64]
42	277 F2:3,	88	BW, FE, FL, FM, FR, FS, FU, FY, LP, SCI	[65]
43	RIL, 178	170	BN, FE, FM, FMAT, FU, FUHML, LI, PH, SCW, SCY, SI, TNMB, TNSB	[66]
44	RIL, 196	104	FE, FL, FM, FU, FS	[67]
45	RIL, 137	280	FS, FT, FBP, PH, NFFB, VR	[68]
46	RIL, -	49	FE, FL, FM, FS, LP, SCW, SCY, SI	[69]
47	RIL, 161	20	VR, FS	[70]
48	RIL, 231	45	FBN, PH	[71]
49	RIL, 188	171	FU, FL, FS, FM, LP, SCW	[72]
50	RIL, 196	104	FS	[73]

**Abbreviation****s:** FE—Fiber elongation; FMAT—Fiber maturity; FSFI—Short fiber index; FUHML—Fiber upper-half mean length; FSCI—Spinning consistency index; FB—Fiber yellowness; MV—Micronaire value; FR—Fiber reflectance, SCY—Seed cotton yield; SY—cotton yield per plant; SI—Seed index; LY—Lint yield; LP—Lint percentage; LI—Lint index; BW—Boll weight; BN—Boll number per plant; LW- Lint weight per boll; HSW—Hundred seed weight; PB—Bolls per plant; NOB—Number of bolls; FN—Fruit branches per plant; NB—Bolls per plant; SW—Seed cotton weight per boll; SCW—Seed cotton weight per boll; Fbbw—Full boll weight; PH—Plant height; FT—Flowering time; TNB—Total number of buds; TNN—Total Number of Nodes; NFFB—Node of the first fruiting branch; SNH—Sympodial Node Height; FSH—First Sympodial Node Height; FBP—Flowering to boll-opening period; FBN—Fruit branch number; TNSB—Total Number of Sympodial Branches; SH—Shoot height; RL—Root length; RSA—Root surface area; RV—Root volume; NRT—Number of root tips; NRB—Number of root branches; RFW—Root fresh weight; LOB—Length of bract; LOP—Length of petal; FTLH—First true leaf height; SLA—Second true leaf area; LFMP—Leaf mid-rib pubescence count; STLH—Second true leaf height; Bla—Bud leaf area; TL—Trichome length; PBS—Percent boll set on second position along sympodia; SFW—Shoot fresh weight; SDW—Shoot dry weight; LOS—Length of staminal column, JI—Jassid injury; FOV—*Fusarium oxysporum* f. sp. *vasinfectum* resistance; CL—Cotton leaf curl disease; VR—*Verticillium dahliae* resistance, CP—Crude protein content; CO—Crude oil content; PA—Palmitic acid content; LA—Linoleic acid content; SA—Stearic acid content; OA—Oleic acid content; NRA—Nitrate reductase activity; Pro—Cottonseed protein content; RWC—Relative water content; SCi—Intercellular CO2 concentration; Oil—Cottonseed oil contents; CSI—Chlorophyll stability index; TTr—Transpiration rate; TPn—Photosynthesis ratio; PC—Proline content; Fcc—Flowering chlorophyll content; TCC—Total chlorophyll content; Bcc—Bud chlorophyll content; Scond—Stomatal conductance; Chla/b—Chlorophyll a/b; Chl a—Chlorophyll a; Chl b—Chlorophyll b.

**Table 2 plants-14-03252-t002:** Characteristics of identified meta-QTLs (MQTLs) across cotton chromosomes, including their genomic positions, confidence intervals, associated traits, and supporting references.

Chr.	MQTL Name	No. of QTLs	Position	CI	Trait	Reference
A01	MQTLchr1-1	10	14.28	0.8	FM, FL, PH, LI, FU, FBN, FE	[39,74]
	MQTLchr1-2	9	27.69	2.24	FM, FU, FL, PH, VR, CP,	-
	MQTLchr1-3	6	44.02	2.05	FM, FBN, PH, TPn, Sci,	[39]
	MQTLchr1-4	5	52.21	0.2	FM, FL, PH	-
A02	MQTLchr2-1	12	3.05	0.49	FS, Fl, FE, PH, SCI, FU, Oil	[39,74]
A03	MQTLchr3-1	17	11.55	1.21	FS, FM, FU, FL, LP, SY, PH, TNS,	[39,47,74]
	MQTLchr3-2	9	44.29	1.53	FL, FM, LP, BN	[39]
	MQTLchr3-3	4	100.46	1.16	LP, FL, NFFB, BN	[75]
	MQTLchr3-4	4	128.9	0.09	FL, FT, NFFB	-
A04	MQTLchr4-1	12	10.33	1.04	[47]	FE, LI, FM, FU, Oil, FUHML, NB, LP
	MQTLchr4-2	7	85.1	1.9	FS, BW	-
A05	MQTLchr5-1	12	2.72	2.6	PH, BW, NFFB, RV, LP, SI, FE, Oil, FBP	[39,47,74]
	MQTLchr5-2	13	12.62	2.6	FBP, FM, FS, LP, FBN, RV, FT, BW, FB	[39,47,74,75,76]
	MQTLchr5-3	11	37.93	2.56	FU, LP, FM, FU, FS, BW, FB	[39,47,74]
	MQTLchr5-4	5	82.83	0.56	RL, PH, FS, RSA, BW	[76]
A06	MQTLchr6-1	9	13.73	1.6	FS, OA, FU, NB, BW	[39,47,74,76]
	MQTLchr6-2	5	47.51	0.79	LI, FMAT, FU, BW, FM	[77]
A07	MQTLchr7-1	15	6.12	0.53	FE, PH, LY, FS, SCY	[39,47,74]
	MQTLchr7-2	7	28.99	1.69	SCY, FBN, FM, FE, LP, FU	[39,74,75]
A08	MQTLchr8-1	22	32.34	1.52	FM, FS, FL, FU	[39,74]
	MQTLchr8-2	8	48.66	1.51	LY, FS, SI, FM	[74,75,77]
	MQTLchr8-3	10	52.88	1.48	VR, FS, SI, LY	[74]
	MQTLchr8-4	12	84.01	1.52	FM, LP, FB, FM, SI	-
A09	MQTLchr9-1	18	2.05	0.42	FS, PH, FL, FU, FBP, VR	[39,74,75]
	MQTLchr9-2	14	18.2	0.4	FU, PH, FS, SI, FSCI, FU, FM, SCY	[39,74]
	MQTLchr9-3	7	32.08	1.33	SI, SCW, Bcc, FT, FS, LP	[74]
	MQTLchr9-4	8	46.44	0.73	FU, Bcc, SCW, FM	-
A10	MQTLchr10-1	10	16.98	3.2	SCW, FMAT, LP, FMIC, FE, FR, PH, NB	[39,74]
	MQTLchr10-2	5	36.16	0.36	NB, SCY, LP, FU, FS	[74]
	MQTLchr10-3	5	58.17	0.76	FM, PA, FT	[77]
A11	MQTLchr11-1	14	0.01	0.06	CL, PB, PH, FE, LP, FL, SI, FB, FM	[39,47,74,77]
	MQTLchr11-2	10	5.35	0.7	FU, FE, Cl, FB, FL, PH, PB	[47,74,77]
	MQTLchr11-3	8	13.21	1.71	FU, SI, FB, FL, FE, PH, PB	[47,74,76,77]
	MQTLchr11-4	5	47.97	1.77	NFFB, FU, FL, TCi, PH	[75,77]
A12	MQTLchr12-1	11	18.94	2.37	STr, SA, FU, TTr, LP, FS, BW, FE, RWC	[74,77]
	MQTLchr12-2	5	51.58	0.75	FU, FBN, LP, RWC	[74,77]
	MQTLchr12-3	7	71.65	0.85	FBP, TNMB, CO, OA, CP	[75]
A13	MQTLchr13-1	14	10.34	1.26	FL, NRT, RL, NFFB, PH, RSA, BW, LP, FE, NRF, PB, FU	[39,74,76]
	MQTLchr13-2	9	38.38	1.51	FE, RSA, NFBB, RL, PH, TNSB	[74]
D02	MQTLchr14-1	23	1.02	0.38	BW, LP, FMIC, FMAT, FU, PH, LY, SCY, PH, SI, FE, FR, FS, LI, RL, FB	[39,47,74,76,77]
	MQTLchr14-2	8	16.99	0.67	FBP, NRF, FU, PH, FM	[39,47,74,76]
	MQTLchr14-3	10	39.74	0.43	FU, LA, FT, CP, FBP, BW, NFFB, FS, FOV	-
	MQTLchr14-4	7	55.04	0.72	FBN, STr, VR, FS, PH, FM	[39]
D01	MQTLchr15-2	19	3.19	0.67	MV, NFFB, FE, SI, SH, FL, SDW, RFW, PH, FUHML, NFFB, LI, SFW	[39,47,74]
	MQTLchr15-1	9	51.62	1.03	FT, NFB, STLH, MIC, PH, Oil	[47,74,77]
D03	MQTLchr17-1	15	86.9	0.21	FT, NFFB, SI, FM, PH	[77]
	MQTLchr17-2	16	95.88	0.86	FT, NFFB, PH, FBP, SI	-
	MQTLchr17-3	7	67.61	0.49	FS, NFFB	[39]
D13	MQTLchr18-1	18	13.03	1.47	BW, FE, FL, SI, LOS, OA, BS, NB, LA	[39,74]
D05	MQTLchr19-1	9	2.06	0.28	FL, FS, FU, BW, BS, SI, PH, CL, SW	[39,47,74,76]
	MQTLchr19-2	14	19.75	0.41	FL, FS, PH, FU, BW, PH, LP, BS	[39,47,74]
	MQTLchr19-3	8	31.53	0.54	BS, FM, TNSB, FS, LP, FOV, SI, LI	-
D10	MQTLchr20-1	10	2.01	0.42	FL, FS, BS, FE, FBN, LY, NFFB	[39,47,74,76]
	MQTLchr20-2	10	28.59	0.56	FBN, SCY, FS, BN, FS, PH	[39]
	MQTLchr20-3	10	47.65	0.49	PH, FM, FS, FU, PH, LI, FE	-
	MQTLchr20-4	6	67.46	0.35	FE, FS, SI, PH, BN	[75]
D11	MQTL21-1	11	0.47	0.23	CL, FM, FL, FMAT, LP	[74,77]
	MQTLchr21-2	6	39.76	0.24	FL, FOV, BW, FMAT, FBN	-
	MQTL21-3	5	74.38	0.08	FM, FBN, FL	[39,77]
	MQTLchr21-4	7	83.99	0.25	FM, PH, FBN, SA, FL, NFFB	-
D04	MQTLchr22-1	7	11.88	1.72	NFFB, FU, FS, FB, FT, FM	[39,74,77]
	MQTLchr22-2	6	53.94	1.27	LP, FM, FS, HSW, SI, FS	[74]
	MQTLchr22-3	6	87.17	0.06	FS, Sci, CP, Tci, Tcond, FE	-
D09	MQTLchr23-1	11	41.85	0.37	VR, FR, FS, NFFB, LI, FU, LP	[39,47,74,76,77]
	MQTLchr23-2	8	56.03	1.56	CO, LI, FL, NFFB, FL, FS, PH, CP	[39,74,77]
D08	MQTLchr24-1	15	47.24	0.13	FBP, LY, SI, NFFB, LP, SCY, FE, SCW, FN, FU	[74,77]
	MQTLchr24-2	10	64.77	0.31	BW, SI, RV, FS, LP, RL, LA, NFFB	[39,77]
	MQTLchr24-3	9	32.36	0.48	LP, FN, FM, LY, FM, FE, SCW	[74]
D06	MQTLchr25-1	10	75.32	0.42	FS, SCW, FL, FE	[74,75]
	MQTLchr25-2	6	31.88	0.75	FU, FS, SCY, BW, FBP	[74]
	MQTLchr25-3	4	86.68	1.03	FS	
	MQTLchr25-4	14	8.33	2.05	FM, FMIC, FE, FL, Oil, FMAT, PH, FL, FSFI	[39,74,76]
D12	MQTLchr26-1	8	24.97	0.48	FL, SDW, FM, FU, MV, LY, SCI	[47]
	MQTLchr26-2	8	35.1	0.3	FM, SI, FT, PH, FE, FBN, FL	[39,74]
	MQTLchr26-3	6	51.43	1.64	FL, BS, BW, SCY, CO, FU	[76]

## Data Availability

Data is contained within the article and Appendix A.

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
