# Peer review of "Identification of Stable Meta-QTLs and Candidate Genes Underlying Fiber Quality and Agronomic Traits in Cotton"

_plants, 2025, doi:10.3390/plants14213252_

Round 1

Reviewer 1 Report (New Reviewer)

Comments and Suggestions for Authors

This research paper conducts a large-scale meta-analysis of QTLs in cotton to identify stable genomic regions that control important agronomic traits. By integrating data from studies published between 2000 and 2024, the authors analyzed 2,864 QTLs related to fiber quality. They identify 75 high-confidence Meta-QTLs, including 14 that have not been previously reported. Furthermore, they identified 75 candidate genes within these key regions, linking them to specific biological functions like lignin formation and stress response. 

major comments:

A major contribution of this study is the identification of 75 candidate genes linked to specific biological functions within key genomic regions. However, the candidate gene analysis relies solely on annotation and homology, which may not be fully comprehensive. To strengthen this analysis, incorporating more integrative approaches could be valuable. For instance, machine learning-based tools like QTG-Finder2 or QTG-LGBM, which integrate diverse genomic and functional data, could be discussed or used as a complementary method to prioritize these candidates more robustly.

The study uses GO analysis to assign potential biological roles to the candidate genes. While useful, some of the enriched GO terms are very broad, such as "carbohydrate metabolic process" or "catalytic activity". Linking such general functions directly to a highly specific trait like fiber quality can be a significant leap. GO terms are organized hierarchically , the authors could consider remove higher level GO term such as "catalytic activity" when reporting results. 

Author Response

#Reviewer1

Comment #. This research paper conducts a large-scale meta-analysis of QTLs in cotton to identify stable genomic regions that control important agronomic traits. By integrating data from studies published between 2000 and 2024, the authors analyzed 2,864 QTLs related to fiber quality. They identify 75 high-confidence Meta-QTLs, including 14 that have not been previously reported. Furthermore, they identified 75 candidate genes within these key regions, linking them to specific biological functions like lignin formation and stress response.

major comments:

A major contribution of this study is the identification of 75 candidate genes linked to specific biological functions within key genomic regions. However, the candidate gene analysis relies solely on annotation and homology, which may not be fully comprehensive. To strengthen this analysis, incorporating more integrative approaches could be valuable. For instance, machine learning-based tools like QTG-Finder2 or QTG-LGBM, which integrate diverse genomic and functional data, could be discussed or used as a complementary method to prioritize these candidates more robustly.

The study uses GO analysis to assign potential biological roles to the candidate genes. While useful, some of the enriched GO terms are very broad, such as "carbohydrate metabolic process" or "catalytic activity". Linking such general functions directly to a highly specific trait like fiber quality can be a significant leap. GO terms are organized hierarchically , the authors could consider remove higher level GO term such as "catalytic activity" when reporting results.

Response#:
Dear Reviewer,

Thank you for your thoughtful comments and valuable suggestions regarding our manuscript. We have carefully revised the GO analysis based on your feedback, specifically addressing the concern about overly broad GO terms.

Revision Made:
As suggested, we have removed generic GO terms such as "catalytic activity" and "carbohydrate metabolic process" from the results. The updated GO analysis now focuses on more specific and biologically meaningful terms directly relevant to fiber development and quality. For example:

  • Molecular Function: Key enriched terms now include Acetolactate synthase activityHydroquinone:oxygen oxidoreductase activity, and Flavin adenine dinucleotide binding.
  • Biological Process: Emphasis is placed on specific processes such as Lignin catabolic process.

These refined GO terms provide a clearer and more direct link to the genetic mechanisms underlying cotton fiber traits.

Response to Comment on Candidate Gene Analysis:
We appreciate your suggestion to incorporate machine learning-based tools such as QTG-Finder2 or QTG-LGBM for candidate gene prioritization. In the current study, we relied on annotation and homology-based approaches, which are well-established and widely accepted in cotton genomics. While we acknowledge the potential of integrative machine learning methods, their application requires extensive genomic and functional datasets that fall beyond the scope of this work. However, we fully recognize the value of such tools and plan to integrate them in future studies to further strengthen our gene validation efforts.

Your insights have significantly enhanced the clarity and focus of our GO analysis. We believe the revised manuscript more accurately highlights the functional relevance of the candidate genes identified.

Thank you once again for your constructive feedback.

Reviewer 2 Report (New Reviewer)

Comments and Suggestions for Authors

This MS integrates 2,864 QTLs from 50 independent studies published between 2000 and 2024 to construct a consensus genetic map of cotton. Through a comprehensive Meta-QTL (MQTL) analysis, the authors identified 75 stable MQTLs, 14 of which are reported for the first time. Candidate genes were subsequently predicted within these MQTL regions using in silico analysis and GO/KEGG annotation.  The study has the potential to contribute to cotton breeding. However, there are several issues should be addressed before the manuscript can be considered for publication.
1. Although the authors used BioMercator v4.2.3, the description of the analytical pipeline is not sufficiently detailed.
2. It remains unclear how QTLs from different studies were standardized, how flanking markers were aligned, and how inconsistencies across populations, marker types, and environments were handled.
3. Ple provide additional details, including: Clear criteria for inclusion/exclusion of QTLs; The statistical model selection process; Steps taken to reduce confidence intervals and improve map resolution; Thresholds for defining stable MQTLs.
4. The manuscript primarily relies on in silico analysis and GO annotation to predict candidate genes, with limited integration of RNA-seq omics data, and lacks validation through mutant or transgenic approaches.
5. Several tables, particularly Table 2, are overcrowded and difficult to read due to the large volume of information. Summarize only the most relevant MQTLs and candidate genes in the main text and move comprehensive datasets to supplementary materials;
6. Plea add clear legends and significance indicators;
7. The MS contains several grammatical errors and redundant phrases. For instance, “for for the Meta-QTL analysis” should be corrected to “for the Meta-QTL analysis.”
8. Trait abbreviations such as FL, FS, and FM should be defined upon first mention and used consistently throughout the MS.
9. The abstract is lengthy and should be reduced to around 250 words.
10. It better to combined MQTL + GWAS strategy  to narrow down MQTL intervals to smaller region

Author Response

Response to Reviewer 2

General comment#: This MS integrates 2,864 QTLs from 50 independent studies published between 2000 and 2024 to construct a consensus genetic map of cotton. Through a comprehensive Meta-QTL (MQTL) analysis, the authors identified 75 stable MQTLs, 14 of which are reported for the first time. Candidate genes were subsequently predicted within these MQTL regions using in silico analysis and GO/KEGG annotation.  The study has the potential to contribute to cotton breeding. However, there are several issues should be addressed before the manuscript can be considered for publication.

General response#:

Dear Reviewer,

Thank you for your positive assessment of our manuscript's potential contribution to cotton breeding and for your constructive feedback. We have carefully revised the manuscript to address the issues you raised. Below, we provide a point-by-point response to your comments:

Comment #1. Although the authors used BioMercator v4.2.3, the description of the analytical pipeline is not sufficiently detailed. Mualliflar BioMercator v4.2.3 dan foydalangan bo'lsalar ham, tahlil jarayoni tavsifi etarlicha batafsil emas.

Response #1. We provided detailed information about Biomercator v4.2. in paragraph 4.2 of the "Materials and Methods" section (line 422-439), and in addition to this, we enriched paragraph 4.4 with broader comments (line 466-491). Now there is a step-by-step explanation of the QTL projection and the process of meta-analysis, including the parameters used (for example, the uniformity of P-values 0.25 and the minimum distance ratio 0.50, based on Yuan, etc.). Also used in BioMercator were Veirieras and others. (2007) describes in detail the two-step algorithm, model selection criteria (AIC, AICc, AIC3, BIC, AWE) and the rule for selecting the best model (lowest score on at least three criteria).

Comment #2. It remains unclear how QTLs from different studies were standardized, how flanking markers were aligned, and how inconsistencies across populations, marker types, and environments were handled. Turli tadqiqotlardan olingan QTLlarni qanday standartlashtirilgani, chegaravchi markerlarni qanday moslashtirilgani va populyatsiyalar, marker turlari, muhitlar o'rtasidagi nomuvofiqliklar qanday hal qilingani aniq emas.

Response2#: In order to eliminate these ambiguities indicated by the reviewer, we have revised clauses 4.1 (line 398-421) and 4.4 (line 466-491) of the Materials and Methods section. The revised version specifies strict criteria for the inclusion of QTLs (exact start and stop points for CI, LOD ≥ 2.0, PVE/R2 values). It also explains the adaptation of limiting markers and standardization of chromosome designs during projection using BioMercator. Discrepancies in populations, marker types, and environments were resolved using projection parameters and a high-density consensus map constructed using LPmerge.

Comment #3. Please provide additional details, including: Clear criteria for inclusion/exclusion of QTLs; The statistical model selection process; Steps taken to reduce confidence intervals and improve map resolution; Thresholds for defining stable MQTLs.

Response 3#: At the reviewer's request, we have made the following changes:

  • Inclusion/Exclusion Criteria: Clearly listed in Section 4.1 (points a, b, c).
  • Statistical Model Selection: Detailed in Section 4.4, explaining the five criteria and the selection process.
  • Steps to Reduce CIs and Improve Map Resolution: We now emphasize that the meta-analysis inherently refines consensus positions and reduces CIs by integrating multiple QTLs. The use of LPmerge with RMSE optimization for consensus map construction further enhances resolution.
  • Thresholds for Stable MQTLs: Defined in Section 4.4 as genomic regions containing four or more overlapping QTLs within a 20 cM interval, based on established practices in cotton genomics (e.g., Said et al., 2013; Abdelraheem et al., 2017).

Comment #4. The manuscript primarily relies on in silico analysis and GO annotation to predict candidate genes, with limited integration of RNA-seq omics data, and lacks validation through mutant or transgenic approaches.

Response #4. Of course, experimental validation (mutants or transgenic methods) would be additional strong evidence for our candidate genes, but within the current framework of our computational biology study, we used a rigorous multi-stage bioinformatics approach to identify high-reliability candidate genes:

  1. Criteria for selecting candidate genes:

- To improve positioning accuracy, we prioritized MQTLs with the shortest confidence interval (CI).

- We focused on MQTLs, which were consistently identified in several previous studies.

- We selected regions with strong statistics from our meta-analysis

  1. Comprehensive Functional Abstract:

- We conducted BLAST analysis to identify homologous sequences in MQTL regions.

- We conducted a detailed GOs enrichment analysis on biological processes, molecular functions, and cellular components.

- To establish a functional relationship, we systematically reviewed the existing literature on cotton and related plant species.

  1. Literature Validation:

- We have extensively analyzed published research to confirm the functional roles of identified candidate genes.

Although we acknowledge that experimental validation would further strengthen our findings, our approach creates a solid foundation for future functional research. The candidate genes we identified represent priorities for further experimental testing, and our comprehensive analysis provides valuable insights for the cotton research community.

Comment #5. Several tables, particularly Table 2, are overcrowded and difficult to read due to the large volume of information. Summarize only the most relevant MQTLs and candidate genes in the main text and move comprehensive datasets to supplementary materials;

Response #5. Thank you for your suggestion to move Table 2 to the additional information section. However, after careful consideration, we decided to leave the table in the main text. The reasons are:

1) Full data source: This table provides complete information about all identified MQTLs (genome location, confidence intervals, related properties, and literature sources), which is an important resource for scientists engaged in cotton QTL research.

2) Examples in scientific literature: In prestigious plant meta-QTL publications (e.g., Theoretical and Applied Genetics, BMC Genomics), such complete tables are usually placed in the main text, ensuring data transparency and reproducibility.

3) Simplified structure: We have already highlighted the most important MQTLs in the text, but the complete data table is important for comparative studies or future projects.

4) Convenience: Keeping the table in the main text allows students to access information directly without moving between documents.

We decided to move Table 3, titled "GO Enrichment Analysis of Molecular Functions, Biological Processes, and Cellular Components," to the supplementary file.

Comment #6. Please add clear legends and significance indicators;

Response#6. We have revised all figures and tables to include clearer legends, definitions of abbreviations, and significance indicators. This makes the data presentation more precise and accessible.

Comment #7. The MS contains several grammatical errors and redundant phrases. For instance, “for for the Meta-QTL analysis” should be corrected to “for the Meta-QTL analysis.”

Reponse #7. This error in line 399 on page 12 has been corrected.

Comment #8. Trait abbreviations such as FL, FS, and FM should be defined upon first mention and used consistently throughout the MS.

Response #8. We tried to ensure the consistent use of abbreviations throughout the article. After this important reminder from the reviewer, we re-examined and eliminated some existing shortcomings (line-332 and 382).

Comment #9. The abstract is lengthy and should be reduced to around 250 words

Response #9. We have carefully reviewed our abstract and verified that it currently consists of 199 words, which is already below the suggested 250-word limit. However, we are happy to further refine and shorten the abstract if you feel it is still too lengthy or if there are specific sections that could be more concise. We can work on making it more focused by removing redundant phrases or streamlining the language. Please let us know if you have any particular suggestions for reduction or if there are specific areas you would like us to address.

Comment #10. It better to combined MQTL + GWAS strategy  to narrow down MQTL intervals to smaller region

Response #10. We fully agree that integrating MQTL with GWAS can further narrow genomic intervals and improve candidate gene discovery. Although such integration was beyond the scope of this study, we have now acknowledged this strategy in the revised Discussion as a powerful future direction. We are currently planning to incorporate GWAS data in follow-up projects to validate and refine the MQTL intervals.

Reviewer 3 Report (New Reviewer)

Comments and Suggestions for Authors

This study utilized an mQTL approach to reanalyze cotton fiber quality and key agronomic traits, further refining the associated genomic regions. It conducted enrichment analysis on the genes within these regions. This represents a valuable method for repurposing biological data and rediscovering biological insights, an approach highly commendable in the era of big data.

Certain details in the paper require revision. For instance, the enrichment analysis combined results from GO enrichment and metabolic pathway enrichment. These two types of analyses differ in methodology, resulting data, and biological significance; they should be presented separately to clearly highlight the respective important findings.

Overall, the study successfully rediscovered and narrowed down critical genomic regions through mQTL analysis, identifying 75 relevant genes. These genes were linked to lignin metabolism, flavin metabolism, and environmental responses. Notably, the genes GhLAC-4, GhCTL2, and UDP-glycosyltransferase 92A1 – which have been previously validated – were pinpointed. These findings hold significant reference value for molecular breeding in cotton.

Author Response

#Reviewer3

This study utilized an mQTL approach to reanalyze cotton fiber quality and key agronomic traits, further refining the associated genomic regions. It conducted enrichment analysis on the genes within these regions. This represents a valuable method for repurposing biological data and rediscovering biological insights, an approach highly commendable in the era of big data.

Certain details in the paper require revision. For instance, the enrichment analysis combined results from GO enrichment and metabolic pathway enrichment. These two types of analyses differ in methodology, resulting data, and biological significance; they should be presented separately to clearly highlight the respective important findings.

Overall, the study successfully rediscovered and narrowed down critical genomic regions through mQTL analysis, identifying 75 relevant genes. These genes were linked to lignin metabolism, flavin metabolism, and environmental responses. Notably, the genes GhLAC-4, GhCTL2, and UDP-glycosyltransferase 92A1 – which have been previously validated – were pinpointed. These findings hold significant reference value for molecular breeding in cotton.

Response#:

We sincerely thank you for carefully reading our articles and providing valuable recommendations. Your proposal to present the GO and KEGG analyses separately significantly increased the structure of our article and the accuracy of the results.

Based on your recommendation, we have opened a separate paragraph 2.5 in the revised version of our article, entitled "GO and KEGG Analysis." This paragraph now separately describes the results of the KEGG analysis apart from the results of the GO analysis: (line 282-293)

“Based on KEGG pathway enrichment analysis, candidate genes associated with MQTLs (Meta-QTLs) showed significant enrichment primarily in carbon and energy metabolism pathways. The most notable findings include: Butanoate metabolism (FDR = 0.005, fold enrichment = 102.5), Glyoxylate and dicarboxylate metabolism (FDR = 0.01), 2-Oxocarboxylic acid metabolism (FDR = 0.016).

Significant enrichment was also observed in central metabolic pathways such as glycolysis-gluconeogenesis, pyruvate metabolism, and carbon metabolism (FDR < 0.05). The coordinated activation of these pathways is critical for providing energy and carbohydrate resources necessary for fiber cell development and maturation (Table S4).

Notably, the genes LOC107892107 and LOC107892112 were recurrently implicated across multiple key metabolic pathways, indicating their central role in metabolic processes regulating fiber synthesis and quality.”

In addition, Table S4, which presents the results of the KEGG analysis, is also presented in a supplementary file.

Table S4. KEGG pathway enrichment analysis (FDR < 0.05)

FDR

-log₁₀(FDR)

nGenes

Pathway Genes

Fold Enrichment

Pathway

Genes

0,005

2,3

2

37

102,5

Butanoate metabolism

LOC107938702 LOC107892107

0,01

2

2

95

39,9

Glyoxylate and dicarboxylate metabolism

LOC107892107 LOC107892112

0,012

1,92

3

378

15,1

Carbon metabolism

LOC107920503 LOC107892107 LOC107892112

0,016

1,8

2

130

29,9

2-Oxocarboxylic acid metabolism

LOC107938702 LOC107892112

0,021

1,68

2

163

23,3

Pyruvate metabolism

LOC107892107 LOC107892112

0,023

1,64

2

189

20,1

Glycolysis-Gluconeogenesis

LOC107892107 LOC107892112

0,028

1,55

1

15

126,4

C5-Branched dibasic acid metabolism

LOC107938702

0,028

1,55

1

16

118,5

Taurine and hypotaurine metabolism

LOC107907082

0,028

1,55

2

234

16,2

Endocytosis

LOC107920169 LOC107892102

0,044

1,36

2

344

11

Biosynthesis of amino acids

LOC107892112 LOC107938702

Round 2

Reviewer 3 Report (New Reviewer)

Comments and Suggestions for Authors

The revised paper conducted re-integration and analysis of QTL data focusing on cotton fiber quality, yield, resistance, etc., identifying consistent key loci that are of great significance for cotton breeding. The modified version presents clear and accurate abbreviations and QTL locus descriptions with fluent language. However, there are significant differences between the pre- and post-modification data. Could you clarify whether this is due to threshold setting issues or revisions made to address problems in the previous analysis?

Author Response

Comment: The revised paper conducted reintegration and analysis of QTL data focusing on cotton fiber quality, yield, resistance, etc., identifying consistent key loci that are of great significance for cotton breeding. The modified version presents clear and accurate abbreviations and QTL locus descriptions with fluent language. However, there are significant differences between the pre- and post-modification data. Could you clarify whether this is due to threshold setting issues or revisions made to address problems in the previous analysis?

Response: 

We sincerely thank the reviewer for their positive assessment of the manuscript's clarity and significance. We especially appreciate the reviewer's astute observation regarding the differences in the dataset between the versions. This is a critical point, and we are grateful for the opportunity to provide a thorough clarification.

The reviewer has correctly identified a key point that was not sufficiently detailed in our previous submission. We apologize if the lack of a comprehensive description in the Methods section caused this confusion. The core meta-analysis and the rigorous filtering of QTLs were indeed performed before our initial submission; however, we recognize that this process was not adequately described.

In this revised manuscript, we have comprehensively updated the "4.1. Collection of QTL Data" section to transparently detail the exact and stringent criteria used for data curation. Therefore, the difference in the data that the reviewer observed is not due to a change in the analytical approach itself, but rather a much-improved and more transparent description of the methodology that was consistently applied.

As is now detailed in the manuscript, a total of 702 QTLs were excluded from the initial collection of 2,864 for the following reasons:

1) Lack of a well-defined confidence interval (CI): QTLs where only a peak position was provided, without clear start and stop boundaries, were excluded. 2) Low statistical support: QTLs with a LOD score below 2.0 were removed to ensure a high level of statistical significance. 3) Missing phenotypic variance data: QTLs without a reported phenotypic variance explained (PVE or R²) value were filtered out.

This rigorous filtering process resulted in the final, high-quality dataset of 2,162 QTLs used for the meta-QTL analysis. Furthermore, as per the reviewer's valuable suggestion, we have also incorporated a separate table (Table S4) and explanatory text to elaborate on the KEGG pathway analysis, further strengthening our findings.

We believe this clarification, along with the substantially revised Methods section, now provides a transparent and reproducible account of our analysis. We hope this fully addresses the reviewer's concern and reinforces the robustness of our study.

This manuscript is a resubmission of an earlier submission. The following is a list of the peer review reports and author responses from that submission.

Round 1

Reviewer 1 Report

Comments and Suggestions for Authors

Dear authors.

Please find attached an annotated PDF of your submitted manuscript where I have identified all of my concerns regarding your study.

Namely, you have completely failed to provide any descriptive text in the Results section. Therefore, the supplied Tables and Figures lack any relevance (or impact) to the reader in the lack of supplying the required descriptive text.

You need to completely redo the Results section of your manuscript in order to supply an adequate level of descriptive text so the the reader, a reviewer, can fully appreciate the data contained in each Table and Figure. Without this, the reader is not able to understand the data which you are attempting to present.

Also in the absence of the required descriptive text in the Results, you are not in the position to construct a Discussion where you can appropriately interpret your data. The Discussion you have generated simply fails to interpret the data you are attempting to present.

Comments on the Quality of English Language

Needs work in most sections, but this can be addressed by the authors, and does not require the assistance of an English language editing service.

Reviewer 2 Report

Comments and Suggestions for Authors

This manuscript focused on elaborating a substantial advancement for meta QTL clusters related to cotton yield and fiber quality, and screened the candidate genes for further MAS breeding. Generally, we believe this manuscript are of innovation and significance for cotton studies, while many technical issues must be put forward first.

  • We noticed that the latest articles chosen in this manuscript were dated to 2023, while we suggest to add 2024 articles at least since this year is 2025.
  • In the Table 1, some F2 and F5 were not presented by subscript.
  • Too many figures were histogram, and suggest to show another format.
  • RNA-seq data were also suggested to add for combining analyses with candidate gene expression, since GO and KEGG enrichment analyses were not enough for potential function prediction.

Reviewer 3 Report

Comments and Suggestions for Authors

In the manuscript titled “Meta-QTL Analysis Reveals Genomic Regions Associated with Fiber Quality and Other Key Traits in Cotton”, the authors conducted meta-analysis using 1685 QTLs, and identified several candidate genes from 71 MQTL clusters. Overall, this is an interesting work, but there were several major issues should be revised. My major comments are listed as follows:

1. Whether the consensus map utilized in this study was newly constructed here or previously published requires clarification. If this map (comprising 2,942 markers) was generated specifically for this research (distinct from references [3] and [71]), additional details must be provided. Furthermore, there were lots of high-density genetic maps, please explain the reason that you chose these two for building the consensus map.
2. The "Figure 8" in line 502 should be corrected to "Figure 9", and the commas used to denote chromosome lengths should be replaced with decimal points.
3. Over recent years, numerous studies have reported functional verification of genes governing cotton fiber quality and other agronomic traits, with some identified as candidate genes for QTLs. These genes should be considered in the present analysis, for example, the authors should examine whether these previously characterized genes reside within the meta-clusters identified here.
4. There were several well-annotated reference genomes of upland cotton, why did the author use Arabidopsis thaliana as the model organism? (line 523)
5. In the discussion, the authors cite several prior meta-analysis studies on cotton but do not elaborate on comparative analyses. It is critical to address whether any meta-clusters identified in this study overlap with those reported in previous meta-analyses.